# Single and Combined Abiotic Stress in Maize Root Morphology

**DOI:** 10.3390/plants10010005

**Published:** 2020-12-23

**Authors:** Rosa Vescio, Maria Rosa Abenavoli, Agostino Sorgonà

**Affiliations:** Dipartimento Agraria, Università “Mediterranea” di Reggio Calabria, Feo di Vito, 89122 Reggio Calabria (RC), Italy; rosa.vescio@unirc.it (R.V.); mrabenavoli@unirc.it (M.R.A.)

**Keywords:** combined stresses, drought stress, heat stress, maize, root morphology, root types

## Abstract

Plants are continually exposed to multiple stresses, which co-occur in nature, and the net effects are frequently more nonadditive (i.e., synergistic or antagonistic), suggesting “unique” responses with respect to that of the individual stress. Further, plant stress responses are not uniform, showing a high spatial and temporal variability among and along the different organs. In this respect, the present work investigated the morphological responses of different root types (seminal, seminal lateral, primary and primary lateral) of maize plants exposed to single (drought and heat) and combined stress (drought + heat). Data were evaluated by a specific root image analysis system (WinRHIZO) and analyzed by uni- and multivariate statistical analyses. The results indicated that primary roots and their laterals were the types more sensitive to the single and combined stresses, while the seminal laterals specifically responded to the combined only. Further, antagonistic and synergistic effects were observed for the specific traits in the primary and their laterals and in the seminal lateral roots in response to the combined stress. These results suggested that the maize root system modified specific root types and traits to deal with different stressful environmental conditions, highlighting that the adaptation strategy to the combined stress may be different from that of the individual ones. The knowledge of “unique or shared” responses of plants to multiple stress can be utilized to develop varieties with broad-spectrum stress tolerance.

## 1. Introduction

The European climate change scenarios will be characterized by extreme temperature, heat waves and warmer days, along with dry days and droughts, especially in Southern Europe [1]. Emerging evidence indicates that these climate change-related events will negatively impact plant/crop/forest productivity in both natural and agro-ecosystems [2,3,4,5,6,7]. To date, the effects and the plant responses to each stressor have been extensively studied at both the morphophysiological and molecular levels. For example, both drought and heat stress reduced the photosynthetic activity, modified the oxidative metabolism, inducing membrane instability [8], changed the phosphoproteome [9] and, consequently, affected the grain yield and quality in maize [10]. However, under field conditions, these abiotic stresses co-occur concurrently during the plant life cycle, stimulating fine-tuned and early prompted plant responses to allocate resources efficiently for adaptation to the coexistent threats. Recent studies uncovered that plants evoke a “unique response” to drought and heat combined stresses [11,12,13], suggesting that their effects are mostly nonadditive (i.e., synergistic and antagonistic) and, therefore, cannot be predicted through single-stressor results. Although few studies have been performed so far, the impacts of the combined drought and heat stress on maize growth, development and yield production and quality have been pointed out [9,14,15,16,17]. In particular, the maize responses to the combined abiotic stress were mainly focused on the aerial part of the plant and its reproductive organs, probably for breeding aims. To the best of our knowledge, no information on the maize root system responses to the combined drought and heat stress are available. This knowledge is even more interesting considering that single maize root types, such as the embryonic roots (primary and seminal) and the postembryonic roots (nodal and lateral) [18], differently respond to environmental cues and could be a “source” of stress adaptation. For example, different responses among root types to drought stress [19,20], allelochemicals [21,22], P deficiency [23] and to combined N deficiency/drought stress [24] were already reported. In this respect, a microcosm experiment was set up for studying several growth and morphological parameters of different root types in response to drought (30% of field capacity) and heat stress (32 °C air temperature) and their combination in maize plants. Further, the additive, synergic and antagonistic effects in combined drought and heat stress were also evaluated.

Considering the cooperation or the antagonism mechanisms of diverse root traits or phenes for root phenotype adaptations in diverse environments [25,26,27], the use of the multivariate approach for identifying the root architecture strategy in terms of functional traits and mechanisms, which operate independently or jointly, was raised [28]. In this respect, the maize root architecture responses to the single and combined stresses were evaluated with the sparse Partial Least Squares-Discriminant Analysis (sPLS-DA), a novel multivariate approach, which, differently to the unsupervised methods (i.e., principal component analysis, the PCA) pointed out very satisfying predictive performances being able to select informative variables [29].

Maize (*Zea mays* L.) is a major cereal crop and food for both humans and animals, widely used as resource for industrial use and for bio-energy production worldwide. It is highly productive under water suitable by irrigation, but water scarcity, high temperature and their combination as observed in semiarid environments caused a reduction on its yield and quality. Hence, the maize tolerance improvement to drought, heat and their combination has become a challenge for the breeding programs [30].

## 2. Results

### 2.1. Univariate Analysis of the Root Morphological Data

One-way ANOVA revealed that each stress (H and D), alone or in combination, significantly affected plant growth in terms of both fresh and dry weight. In particular, the combined stress reduced the fresh weight more than heat ones in respect to the control but similarly to drought stress. By contrast, all the stresses diminished the plant dry weight to a similar extent (Figure 1).

Figure 2 showed that the single root types of maize seedlings were also modified by single and combined abiotic stress.

Out of the eleven traits of the primary roots, eight were significantly modified by stresses in comparison to the control (Table 1). The drought stress significantly increased the root length ratio (RLR), root mass ratio (RMR), fineness, tissue density and branching zone but reduced the branching density; conversely, the RLR and branching zone were not affected by heat stress, which, in turn, weakly increased the fineness and tissue density and reduced the branching density (Table 1). The combined stress pattern was similar to the drought stress but with a sharp increase of the dry weight and, consequently, of both the RMR and tissue density and a reduction of the branching density (Table 1).

The morphology of the seminal roots was lesser affected and not by all the stresses (Table 2). In particular, the drought stress reduced the fresh weight but increased the fineness only; the heat stress increased the fineness but, differently from the drought stress, raised the branching density (Table 2). Conversely to the single stress, the combined stress reduced the fresh weight only with respect to the control (Table 2).

Six traits of the primary lateral roots were differentially modified by the stress (Table 3). The RLR and fineness were increased by drought stress in comparison with the control, while the heat stress increased the fresh weight, length, surface area and RLR but not the fineness. The combined stress pointed out a significant and marked modification of the primary lateral root morphology increasing by 84%, 56%, 124%, 43% and 57% the length, surface area, RLR, fineness and average length, respectively (Table 3).

The seminal lateral roots were affected by the stress in eight out of ten traits (Table 4). However, the single stress lesser influenced these root types, which reduced the RMR (drought stress) and the dry weight and surface area (heat stress) in comparison to the control (Table 4). Conversely, the combined stress strongly affected the seminal lateral roots, reducing the dry weight, length, surface area and RMR and increasing the fineness (Table 4).

### 2.2. Additive, Synergistic and Antagonistic Effect of Combined Stress

In order to evaluate the nonadditive effects on the root traits significantly modified by the combined stress, we used the multiple risk model [31] that eluded the overinflated response estimated by a simple additive model.

The response to the combined stress in maize root types were the result of the additive, synergistic and antagonistic effects of the single stress depending on the root types and traits (Figure 3). Indeed, the significant increase in the fineness and the decrease in the branching density in the primary roots under the combined stress (Table 1) was the result of an antagonistic effect of the single stress, whereas an additive effect was evoked for the enhance of dry weight, RLR, RMR, tissue density and branching zone (Figure 3A). The decrease of the fresh weight in the seminal root, the only trait significantly modified by the combined stress (Table 2), was the result of the additive effect (Figure 3B). A synergistic effect could justify the increase of the length, surface area and the RLR in the primary lateral roots exposed to the combined stress, while an additive effect was evoked for the fineness and average length (Figure 3C). Finally, the decrease of the length of the seminal lateral roots in response to the combined stress was due to a synergistic effect, whereas an additive effect was responsible for the modifications of the dry weight, surface area, RMR and fineness (Figure 3D).

### 2.3. Root Responses to the Single and Combined Stress: A Supervised Analysis with PLS-DA

A Permutational Multivariate Analysis of Variance (PERMANOVA) revealed that all the treatments determined a significant difference (*p* < 0.001) in the root morphology, as reported in Table 5. The pairwise PERMANOVA comparisons suggested that the combined and drought stress, but not the heat ones, triggered significant differences with respect to the control. Furthermore, the drought stress caused a different root morphology in comparison to both the heat and combined stress (Table 5).

In order to select informative and relevant root traits, we used sparse Partial Least Squares-Discriminant Analysis (sPLS-DA), a multivariate method characterized by a very satisfying predictive performance, for the multiclass classification in plant biological studies [29,32].

The performance step for the selection of the number of components suggested that three were enough to sharply reduce the balanced error rate (Figure 4A). Further, the final model obtained by the tuning process pointed out that each component was constituted by one root morphological trait with a Balanced Error Rate (BER) around 0.18 (Figure 4B). The sample plots on the three components permitted to visualize a sharp discrimination among the treatments with 52% of the total explained variability split up by 27%, 14% and 11% for the first, second and third components, respectively (Figure 5A,B). In particular, plotting the first two components (component 1 and component 2), the combined and, to a lesser extent, the drought stress were sharply separated from the control by the first component, whereas the second component discriminated against the drought and, at lesser degree, the heat plants with respect to the control (Figure 5A). The addition of the third component permitted to separate the heat plants from the control ones (Figure 5B).

Figure 5C–E showed the selected root traits and relative loading weights for each component, and the color indicated the treatments for which of the selected root traits had a maximal mean loading weight value. In particular, the root traits identified as performants for the sPLS-DA model were the primary lateral RLR in the combined stress, primary root branching density in the control and seminal root branching density in the drought stress for the first, second and third components, respectively (Figure 5C–E).

## 3. Discussion

### 3.1. Single Stress Determined Different Root Type-Related Morphological Responses

Drought, as well as heat stress, affected the primary root more than the other types. Indeed, differently from the seminal root, the primary improved its length both in term of absolute (LR) and relative values (RLR) but with different intensities: higher in drought than heat plants. The primary root is the very early dominant type in maize seedlings determining the early vigor and, by deepening in the subsoil strata, survival under a water deficit and higher air temperature [20,33]. Further, the rooting depth is a phene very interesting, as it confers drought tolerance/resistance in several plant species such as rice [34] and wheat [35], improving the subsoil water capture. In order to evaluate the “morphological pattern” determining the increase in the RLR, a trait that, better than the absolute length, is related to the plant’s potential for water and nutrient acquisition under stress conditions [36], we estimated its “morphological components”—that is, RMR, fineness and tissue density. According to Ryser and Lambers (1995) [37], the higher RLR could be due to an increase of the RMR and/or fineness and/or decrease of the tissue density, as explained by the Equation (1). The results indicated that the increase of RLR in the primary root, under both single stress conditions, was due to a concomitant enhancement of the biomass allocation (RMR) and fineness, accompanied by a higher root tissue density (RTD) trait positively correlated with the degree of lignification and cell wall thickness [38,39,40]. Hence, the maize seedlings under both drought and heat stress, increased the length of the primary root, which appeared finer and, at the same time, thick, useful traits to penetrate hard soil layers under water stress [41]. Note that the RLR of the primary root under the drought stress was also higher than the heat ones, and this difference was due to a higher fineness rather than the biomass allocation variation (Table 1). Extensive transcriptomic and proteomic studies revealed specific transcripts and proteins related with cell wall extension properties in the primary root of maize seedlings exposed to the water stress (see References [42,43]). Therefore, according to our results, the drought stress could induce a different molecular mechanism with respect to the heat one, which differently regulated the elongation of the primary roots, as observed in this study.

The seminal root axes were the lesser modified by the single stresses, which determined an increase in their fineness only (Table 2). The maize seminal roots responded to the drought and heat stress by reducing their emergence angle and length, which resulted in the soil deepening [20,24]. These results were not observed in this study, probably, for the pot volume that limited the soil exploration.

Besides the primary and seminal root axes, the laterals, as postembryonic roots arising from these axes, played an important role in water acquisition from the soil, allowing an improvement of the soil exploration due to their higher surface-to-volume ratio [44]. Differently to the seminal lateral roots, the primary laterals were more modified by both single stresses: the drought-stressed plants pointed out similar lateral lengths to the heat plants but with a higher fineness (Table 3 and Table 4). Again, the trait “fineness” was differently regulated by drought with respect to heat stress, allowing a higher surface soil contact of the laterals, fundamental for water uptake. Note that, although the length of the primary lateral roots was increased, the branching density was decreased in both stressed plants. This reduction, together with an increase of length of the primary lateral roots, constituted an important root “phenotypic pattern”, which improved the drought resistance in maize plants by reducing both the intraplant competition for the photosynthates and the capturing of mobile soil resources such as water [19].

### 3.2. Combined Stress Caused Different Root Type-Related Morphological Response with Respect to the Single Stress with Nonadditive Effects

The recent studies on combined stresses, such as drought and heat, were focused on the morpho-physiological and molecular responses of plant aerial traits such as yield and quality [45,46], plant growth [47], foliar chemistry [48] and leaf physiology [14,49]. Conversely, very few studies have focused on the root system responses to combined stress, and no information is available for the different root types. Here, for the first time, the responses of the single root types of maize seedlings to the combined stress were reported. It induced a similar morphological pattern to the single stresses, resembling those of drought and heat stress for the primary and seminal root axes and those of heat for the primary lateral roots only. Conversely, the morphology of the seminal lateral roots of the combined stress was completely different from that of the single stresses: length and biomass were sharply inhibited. Probably, the water scarcity exacerbated by the heat rendered not useful the exploration and resource exploitation of the topsoil strata; hence, the maize plants engaged their internal resource towards the root classes, such as the primary and its laterals, mainly localized in the subsoil to reach the water reserve.

Analyzing the pattern of the combined stress on the root traits is interesting to understand their additive (equal to the sum of the single-stress effects), synergistic (higher than expected) or antagonistic (lower than expected) effects. Besides useful information on the morphological pattern, these could provide a hypothesis on the signaling pathways and molecular mechanisms underlying the plant strategy in the presence of simultaneous stress. For example, the synergistic effect of NaCl and ABA in *Arabidopsis thaliana* was induced by the expression of *Responsive-to-Dehydration 29A* (*RD29A*) that cannot be explained by the sum of responses to the single stresses [50]. Furthermore, the antagonistic effect of drought and insect herbivory could be explained by the synergistic interactions between JA and ABA signaling [51]. In our study, the combined stress produced mostly additive effects, but synergistic and antagonistic effects for specific trait and root types were also observed (Figure 3), suggesting, for these latter, unique molecular and signaling interaction mechanisms. These results are discussed below for their ecological role and plant fitness in an environment characterized by co-occurring drought and heat stress. Differently to the seminal and their lateral roots, the primary lateral root traits were mostly synergized, whereas those related to the primary root axes were antagonized (Figure 3). Why were nonadditive effects (synergistic and antagonistic ones) observed in the primary and their lateral respect to the seminal roots? Their different locations within the soil environment that cause diverse efficiency for resource acquisition could be hypothesized. Indeed, primary and their lateral roots are placed in the water-rich subsoil strata, making these root types more important under water scarcity due to lesser rainfall/irrigation, further aggravated by simultaneous heat stress. Root deepening during drought stress was revealed in maize plants [20,24,35]. However, our results pointed out that combined stress antagonized the fineness and the branching density of the primary roots (Figure 3A). Probably, the thicker primary roots could be more useful for deeper penetration in the compacted soil caused by both drought and heat stress [52,53]. At the same time, the reduction of the branching density of the primary root axes, along with the root deepening, are useful strategies for root adaptation in water scarcity environments, as observed by [54]. Next to the antagonistic effects in the primary roots, the synergistic effects were observed on the length, surface area and RLR traits of the primary lateral roots (Figure 3C). Sustained by the soil penetration of the thicker primary roots, the longer laterals with higher surface contacts with the soil could increase the exploration of the subsoil strata characterized by higher soil moisture under combined drought and heat stress.

### 3.3. Primary Lateral RLR and Primary and Seminal Root Branching Density Discriminated the Root Phenotypes in Drought and Heat Stress and Their Combination

Currently, for understanding the root strategy for the plant adaptation to abiotic stress, the multivariate analysis was applied. This suitable method allowed to identify the efficient and meaningful “multi-trait classifiers” of the root systems [28] and the detection of the functional root traits and mechanisms, which operate independently or jointly. For example, the principal component analysis (PCA) permitted to identify the root ideotypes in drought stress for peanut landraces [55], sugar beet genotypes [56], bean landraces [57] and soybean genotypes [58]. Conversely, the sparse PLS-DA, a supervised technique, made efficient the trait selection and dimension reduction simultaneously by imposing sparsity to the solution [59], making it a novel approach to investigate high-dimensional and redundant root data.

The sPLS-DA clearly separated the root phenotypes of the combined and, to a lesser degree, the drought stress from that of the control by Component 1 (Figure 5A). This component showed high positive loading for the RLR of the primary lateral roots, and the combined stress was the group for which the selected variable had a maximal mean value. Hence, the root phenotypes of the combined stress were different from that of the control plants, as firstly suggested by the PERMANOVA analysis, and the RLR of the primary lateral roots was the triggering root trait, as pointed out from the sPLS-DA. As already described, this trait, which expresses the relative investment of the plant in the root length, was strictly related to the soil resource availability and, hence, for their capture [37,60], depended on the nutrient availability [61,62] and drought stress [56]. Components 2 and 3 separated the root phenotypes of the drought stress plants from that of the control and heat ones, respectively (Figure 5A,B). The primary and seminal root branching densities were the root traits with a maximal negative mean value for the control and drought groups, respectively, confirming that the interbranch length could play a fundamental role in the drought stress condition [19,54].

## 4. Materials and Methods

### 4.1. Plant Material and Growth Condition and Treatment

The experiments were conducted at the University “Mediterranea” of Reggio Calabria, Italy. Maize seeds (*Zea mays* L.) (genotype KXB7554, provided by KWS Italia) were surface-sterilized with 20% NaClO for 20 min, rinsed and then soaked in aerated deionized water at room temperature for 36h. Afterwards, five seeds were sown in each of sixteen sterilized pots (16-cm diameter × 12 cm height), which were filled with a sand:soil mixture (70:30 *v*/*v*). The soil physicochemical values were reported in Gelsomino et al. [63]. Then, the pots were randomly placed in the growth chamber at 25 °C, 70% relative humidity and 350 μmol m^−2^ s^−1^ of the photosynthetic photons flux density at plants’ height (LI-190SA quantum sensor, Li-Cor, Lincoln, NE, USA) with a 14-h photoperiod. The planted pots received, for two weeks, 200 mL of tap water every four days, necessary to compensate the water losses by evapotranspiration, as suggested by the preliminary trials. After twelve days from seeding, five seedlings were thinned to one for each pot.

### 4.2. Treatments

From the third week, the planting pots were subjected to stresses. In particular, eight pots were transferred to a second growth chamber with the same environmental conditions, except for the temperature set to 32 °C (heat condition, H), whereas the remaining eight were left in the previous growth chamber at 25 °C. In four pots of both growth chambers were imposed drought stress (D) by water withholding until reaching the theoretical fraction of 30% field capacity (FC) measured by the gravimetric method. To the remaining pots, conversely, were ensured well-watered conditions in order to maintain the theoretical fraction of 80% FC. The desired percentage of FC was maintained by the daily surface addition of water. Overall, the following treatments were imposed: drought (D) (25° and water at 30% FC), heat (H) (32 °C and water at 80% FC), combined (Comb) (32 °C and water at 30% FC) and control (Con) (25° and water at 80% FC).

### 4.3. Morphological Root Analysis

After 7 days of treatments, the seedlings were harvested and separated into shoot and root. The shoot fresh (ShFW, g) was measured, and then, the shoot dry weight (ShDW, g) was determined after oven-drying at 70 °C for 48 h.

After gently removing the adhering substrate, the root system was washed by tap water, paper-blotted and then stained with 0.1% toluidine blue solution for 5 min. Afterwards, it was divided into primary and seminal roots with their laterals and scanned at a resolution of 300 dpi (WinRhizo STD 1600, Instruments Régent Inc., Québec, QC G1V 1V4, Canada) with the WinRHIZO image analysis (WinRhizo STD 1600, Instruments Régent Inc., Canada). Then, the length (L, cm), surface area (SA, cm^2^) and volume (V, cm^3^) of the seminal (S) and primary axes (P) and seminal (SL) and primary lateral roots (PL) were measured by WinRhizo Pro v. 4.0 software package (Instruments Régent Inc., Canada). The number of both primary and seminal lateral roots was manually counted from the scanned image. The root and branching zone formation (RZF and BZF, respectively; cm) and the branching density (BD, n cm^−1^) were also measured and calculated, respectively, as reported by Drubvosky et al. [64].

The fresh and dry weights of the primary (PrFW, g and PrDW, g), seminal (SFW, g and SDW, g), primary lateral (PrLFW, g and PrLDW, g) and seminal lateral roots (SLFW, g and SLDW, g) were measured as reported above. The fresh (RFW, g) and dry weights (RDW, g) of root system were calculated as the sum of each root type, and the plant fresh (PFW, g) and dry weight (PDW, g) were calculated by the sum of the ShFW and RFW and the ShDW and RDW, respectively.

Finally, based on the above measurements, the root length ratio (RLR, root length/whole plant dry weight, cm g^−1^), root mass ratio (RMR, root dry weight/whole plant dry weight, g g^−1^), root fineness (RF, root length/root volume, cm cm^−3^) and root tissue density (RTD, root dry weight/root volume, g cm^−3^) were calculated for each root type. The functional significance of these root parameters was reported by Ryser [36]. Further, according to Ryser and Lambers [37], the RLR and its “morphological components” (RMR, RF and RTD) are related as follows:
(1)$$\mathrm{RLR}=\mathrm{RMR}\times \frac{\mathrm{RF}}{\mathrm{RTD}}$$

### 4.4. Statistical Analysis

All the experiments were arranged in a randomized complete design with four replicates per treatment.

All the root morphological parameters were firstly analyzed by one-way ANOVA followed by the Tukey’s test to compare the mean values among the treatments (Control, D, H and Comb) at *p* < 0.05.

To determine if the combination of the H and D stress-exerted additive, synergistic or antagonistic impacts on the root traits, we used the Bansal et al. method [65], and, specifically, we compared the observed effects (Ob) to expected additive effects (Ex) for the plants exposed to the Comb treatment. The Ob effect sizes were calculated as the absolute value of:
(2)$$\mathrm{Ob}=\frac{\left(\mathrm{ob}-\bar{x}\mathrm{Con}\right)}{\bar{x}\mathrm{Con}}$$
where ob is the measured trait value for each plant and treatment, and $\bar{x}\mathrm{Con}$ is the mean trait value for the control plants. The Ex additive effect sizes for the Comb treatment were defined in two steps by first determining and then summing the independent effects (Ind) of each treatment. The Ind effect sizes were calculated as the absolute value of:
(3)$$\mathrm{Ind}=\;\frac{\left(\bar{x}\mathrm{stress}-\bar{x}\mathrm{Con}\right)}{\bar{x}\mathrm{Con}}$$
where $\bar{x}\mathrm{stress}$ is the mean trait values from each stress (H and D), and $\bar{x}\mathrm{Con}$ is the mean trait value for the control plants. Then, the Ex additive effect size for the Comb treatment was calculated using a multiplicative risk model, as suggested by Darling et al. [31], i.e., the sum of two Ind effects minus their product. Finally, the Ex additive values for the Comb treatment were compared to the actual Ob additive effects. In particular, we calculated a mean difference (± 95% confidence interval) between the effect sizes of the Ob and Ex for each seedling of the Comb treatment. When Ob-Ex > 0 and the lower 95% confidence limit was greater than zero, the impact from the combination of both stressors was classified as synergistic. Conversely, the effects were antagonistic when the Ob-Ex < 0 and the upper 95% confidence limit was less than zero and additive when the 95% confidence interval crossed the zero line.

Furthermore, we analyzed the effects of the single and combined stresses on the entire dataset of the root morphological parameters using a multivariate approach with R statistical software 3.5 (R Core Team, 2013). First, the differences among treatments were inferred through the PERMANOVA multivariate analysis (999 permutations) using the package *vegan*. Pairwise comparisons among the groups were calculated using a custom script and correcting *p*-values using the False Discovery Rate (FDR) method. In order to identify the root morphological key predictors that could constitute a root strategy among the treatments, we used a preliminary unsupervised (Principal Component Analysis, PCA) followed by the supervised analysis (sparse Projection to Latent Structure-Discriminant Analysis, sPLS-DA) using the package *mixOmics* [66]. The *perf.plsda()* and *tune.splsda()* functions were used to predict the number of latent components (associated loading vectors) and the number of discriminants root traits for the sPLS-DA, respectively.

In particular, the optimal number of components was chosen by the averaged overall and balanced classification error rates with centroid distances over 50 repeats of 5-fold cross-validations (*perf.plsda()*). The optimal number of root traits for each component was then selected by the lowest average balanced classification error rate with centroids after tuning of the sPLS-DA model (*tune.pldsda()*) using the selected number of components and 5-fold cross-validation with 50 repeats. Single samples were showed on a score plot and differentiated by treatments with color and 95% confidence ellipses. Furthermore, the discriminant root traits were plotted according to their contribution weights to Components 1, 2 and 3 of the sPLS-DA and discriminated by treatments with color.

Finally, the Pearson product–moment correlations between the plant fresh and dry weight with the scores of the latent components determined by the sPLS-DA were run for verifying the root strategy for plant adaptation to abiotic stress.

The statistical software was SPSS Inc., V. 10.0, 2002 (SPSS Inc., Evanston, IL, USA).

## 5. Conclusions

The present study pointed out, for the first time, the responses of different root types to combined abiotic stresses in maize seedlings.

The single and combined stresses caused fine variations in the growth and morphology of the single root types in the maize root phenotype. The seminals were the least modified root types, whereas the primary and their lateral roots were stimulated with an increase of the length together with a higher biomass allocation and fineness by the single stress conditions. The combined stress determined similar effects but was associated with a specific inhibition of growth and morphology of the seminal lateral roots. Nonadditive effects (synergistic and antagonistic) were only observed in the primary and their lateral roots under the combined stress, suggesting that single molecular mechanisms could underlie their growth and morphological responses. Further, the results of the sPLS-DA supported the idea that the primary and their lateral roots could be the “root type” with an important role for adaptation to the combined abiotic stress.

## Figures and Tables

**Figure 1 plants-10-00005-f001:**
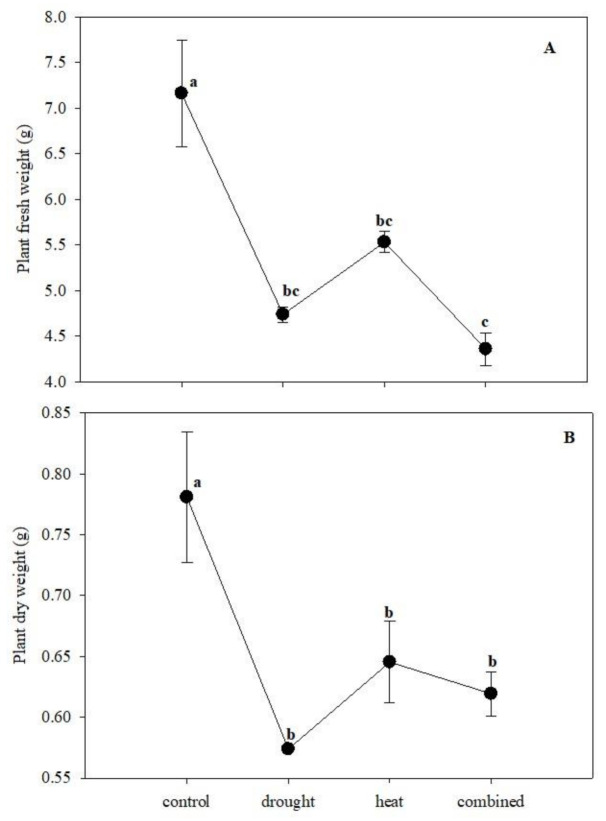
The maize growth in terms of fresh (**A**) and dry weight (**B**) in the presence of single (drought and heat stress) and combined stress. The bars represented the error standard (N = 4). Different letters indicated significant difference among the treatments (*p* < 0.05; Tukey’s test)

**Figure 2 plants-10-00005-f002:**
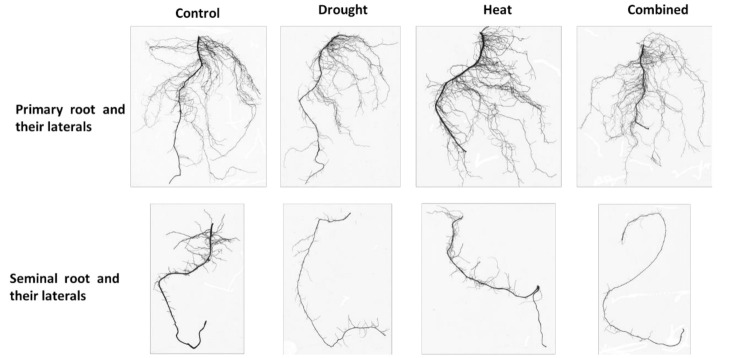
Primary and seminal roots and their laterals of maize seedlings exposed to drought, heat and their combination (Combined).

**Figure 3 plants-10-00005-f003:**
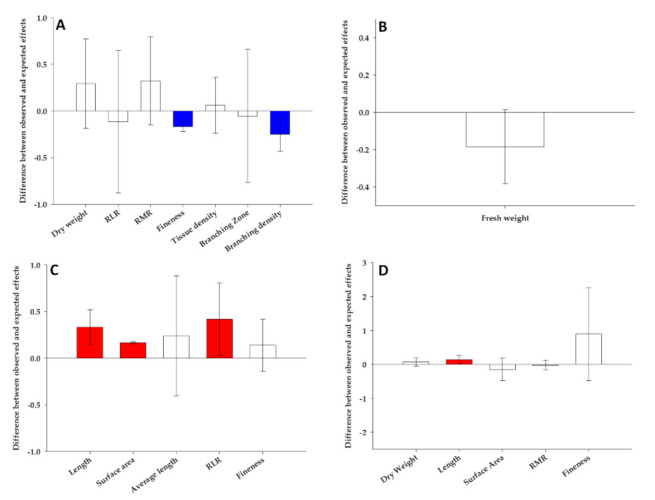
The combined impacts from drought and heat stress on selected traits of root primary (**A**), seminal (**B**), primary lateral (**C**) and seminal lateral (**D**) of maize plants. The combined impact of single stressors was estimated as synergist (red color), additive (white color) or antagonistic (blue color) (greater than, equal to or less than expected effects, respectively, based on single stressor effect sizes). The vertical and error bars represent, respectively, the mean and the 95% confidence intervals of the overall effect size differences between the observed and expected additive effects from combined drought and heat on the root traits of maize plants. The zero line represents the expected additive effects from combined stressors. When the means (and their 95% confidence limits) were higher than or less than the zero line, they were considered synergistic or antagonistic, respectively. RLR: root length ratio and RMR: root mass ratio.

**Figure 4 plants-10-00005-f004:**
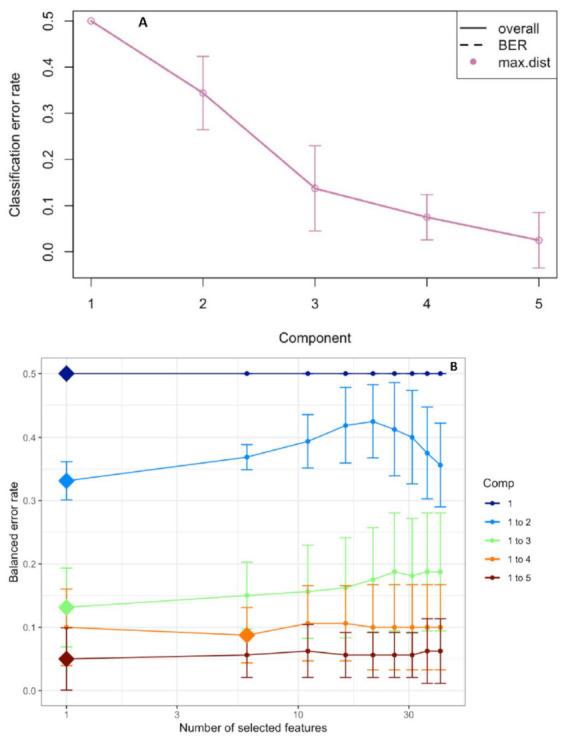
Choosing the number of components in the sparse Partial Least Squares-Discriminant Analysis (sPLS-DA) by the performance test (**A**). Mean classification by the overall and balanced error rate (BER) (5 cross-validations averaged 50 times) for each sPLS-DA component. Choosing the number of root traits for each sPLS-DA component by a tuning test (**B**). Estimated classification balanced error rates for the root morphology dataset (5 cross-validations averaged 50 times) with respect to the number of selected root traits for the sparse exploratory approaches.

**Figure 5 plants-10-00005-f005:**
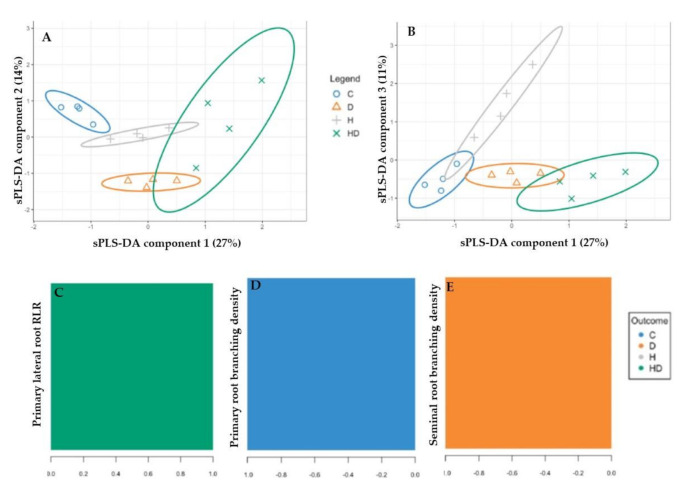
sPLS-DA sample plot for the different components using 95% confidence ellipses. (**A**) Component 1 vs. Component 2 and (**B**) Component 1 vs. Component 3. Contribution plots by loading weights of the root traits selected for each sPLS-DA component: (**C**) Component 1, (**D**) Component 2 and (**E**) Component 3.

**Table 1 plants-10-00005-t001:** Morphology of primary root of maize plants exposed to single (drought and heat) and combined stress (drought + heat).

Category	Parameters	Treatments			
		Control	Drought	Heat	Combined
biometric	Fresh weight (g)	0.128 (0.008) ^a^	0.131 (0.001) ^a^	0.147 (0.001) ^a^	0.162 (0.021) ^a^
	Dry weight (g)	0.0111 (0.0009) ^b^	0.0116 (0.0006) ^b^	0.0131 (0.0003) ^b^	0.0168 (0.0017) ^a^
geometric	Length (cm)	49 (1) ^a^	71 (6) ^a^	57 (4) ^a^	66 (12) ^a^
	Surface area (cm^2^)	9.460 (0.009) ^a^	10.669 (0.896) ^a^	9.723 (0.587) ^a^	10.415 (1.766) ^a^
Length components	RLR (cm g^−1^)	63 (4) ^b^	123 (12) ^a^	83 (2) ^b^	117 (15) ^a^
	RMR (g g^−1^)	0.01381 (0.00016) ^c^	0.01840 (0.00002) ^b^	0.02004 (0.00051) ^b^	0.02702 (0.00205) ^a^
	Fineness (cm cm^−3^)	336 (11) ^c^	531 (11) ^a^	396 (8) ^b^	500 (6) ^a^
	Tissue density (g cm^−3^)	0.075 (0.007) ^c^	0.097 (0.006) ^b^	0.097 (0.003) ^b^	0.117 (0.007) ^a^
branching	Root zone formation (cm)	9 (1) ^a,b^	3 (1) ^b^	9 (3) ^a^	7 (1) ^a,b^
	Branching zone formation (cm)	42 (2) ^b^	65 (5) ^a^	48 (1) ^a,b^	64 (11) ^a^
	Branching density (n cm^−1^)	4.961 (0.101) ^a^	3.389 (0.051) ^c^	4.276 (0.027) ^b^	4.152 (0.284) ^b^

Different letters along the rows indicated significative differences among the means at *p* < 0.05 (test of Tukey). The values within the brackets are the error standards (n = 4). RLR; root length ratio and RMR: root mass ratio.

**Table 2 plants-10-00005-t002:** Morphology of seminal root of maize plants exposed to single (drought and heat) and combined stress (drought + heat).

Category	Parameters	Treatments			
		Control	Drought	Heat	Combined
biometric	Fresh weight (g)	0.285 (0.031) ^a^	0.172 (0.029) ^b^	0.205 (0.043) ^a,b^	0.176 (0.018) ^b^
	Dry weight (g)	0.021 (0.004) ^a^	0.015 (0.003) ^a^	0.014 (0.004) ^a^	0.018 (0.002) ^a^
geometric	Length (cm)	64 (7) ^a^	64 (7) ^a^	63 (8) ^a^	51 (2) ^a^
	Surface area(cm^2^)	11 (1) ^a^	10 (1) ^a^	12 (2) ^a^	10 (1) ^a^
Length components	RLR(cm g^−1^)	82 (13) ^a^	103 (14) ^a^	97 (14) ^a^	83 (1) ^a^
	RMR (g g^−1^)	0.027 (0.002) ^a^	0.023 (0.002) ^a^	0.027 (0.007) ^a^	0.031 (0.003) ^a^
	Fineness (cm cm^−3^)	294 (47) ^b^	463 (21) ^a^	435 (55) ^a^	362 (24) ^a,b^
	Tissue density (g cm^−3^)	0.094 (0.007) ^a,b^	0.064 (0.023) ^b^	0.091 (0.003) ^a,b^	0.124 (0.001) ^a^
branching	Root zone formation (cm)	7 (1) ^a,b^	8 (2) ^a,b^	12 (3) ^a^	6 (1) ^b^
	Branching zone formation (cm)	54 (5) ^a^	55 (6) ^a^	57 (14) ^a^	44 (3) ^a^
	Branching density (n cm^−1^)	2.8 (0.1) ^b^	2.9 (0.1) ^b^	4.5 (0.4) ^a^	3.0 (0.2) ^b^

Different letters along the rows indicated significative differences among the means at *p* < 0.05 (test of Tukey). The values within the brackets are the error standards (n = 4).

**Table 3 plants-10-00005-t003:** Morphology of primary lateral root of maize plants exposed to single (drought and heat) and combined stress (drought + heat).

Category	Parameters	Treatments			
		Control	Drought	Heat	Combined
biometric	Fresh weight (g)	0.212 (0.012) ^b^	0.280 (0.047) ^a,b^	0.315 (0.040) ^a^	0.225 (0.001) ^a,b^
	Dry weight (g)	0.022 (0.004) ^a^	0.016 (0.001) ^a^	0.021 (0.003) ^a^	0.020 (0.001) ^a^
geometric	Length (cm)	557 (8) ^c^	691 (35) ^b,c^	757(82) ^b^	1028 (33) ^a^
	Surface area (cm^2^)	34.6 (0.7) ^b^	37.2 (2.7) ^b^	45.8 (4.8) ^a^	53.3 (0.1) ^a^
	number (n)	208 (16) ^a^	225 (14) ^a^	200 (2) ^a^	247 (27) ^a^
	Average length (cm)	2.860 (0.136) ^b^	3.076 (0.039) ^b^	3.640 (0.384) ^a,b^	4.486 (0.578) ^a^
Length components	RLR (cm g^−1^)	736 (41) ^c^	1195 (63) ^b^	1125 (75) ^b^	1650 (90) ^a^
	RMR (g g^−1^)	0.0324 (0.0079) ^a^	0.0220 (0.0003) ^a^	0.0308 (0.0031) ^a^	0.0330 (0.0016) ^a^
	Fineness (cm cm^−3^)	3257 (44) ^c^	4018 (56) ^b^	3512 (76) ^c^	4670 (288) ^a^
	Tissue density (g cm^−3^)	0.134 (0.027) ^a^	0.093 (0.005) ^a^	0.097 (0.005) ^a^	0.093 (0.006) ^a^

Different letters along the rows indicated significative differences among the means at *p* < 0.05 (test of Tukey). The values within the brackets are the error standards (n = 4).

**Table 4 plants-10-00005-t004:** Morphology of seminal lateral roots of maize plants exposed to single (drought and heat) and combined stress (drought + heat).

Category	Parameters	Treatments			
		Control	Drought	Heat	Combined
biometric	Fresh weight (g)	0.0600 (0.0114) ^a,b^	0.0355 (0.0048) ^b^	0.1323 (0.0459) ^a^	0.0149 (0.0002) ^b^
	Dry weight (g)	0.0051 (0.0010) ^a^	0.0035 (0.0006) ^a,b^	0.0025 (0.0008) ^b^	0.0013 (0.0002) ^b^
geometric	Length (cm)	113 (7) ^a^	104 (1) ^a,b^	105 (13) ^a,b^	79 (14) ^b^
	Surface area (cm^2^)	8.5 (1.3) ^a^	7.5 (0.1) ^a^	3.7 (0.6) ^b^	4.5 (0.9) ^b^
	number (n)	149 (21) ^a,b^	169 (18) ^a,b^	202 (23) ^a^	134 (5) ^b^
	Average length (cm)	0.872 (0.151) ^a^	0.707 (0.081) ^a^	0.563 (0.136) ^a^	0.530 (0.094) ^a^
Length components	RLR (cm g^−1^)	147 (1) ^a,b^	178 (2) ^a^	151 (12) ^a,b^	128 (19) ^b^
	RMR (g g^−1^)	0.0066 (0.0010) ^a^	0.0032 (0.0001) ^b^	0.0041 (0.0015) ^a,b^	0.0021 (0.0003) ^b^
	Fineness (cm cm^−3^)	2728 (48) ^b^	2416 (0) ^b^	3630 (376) ^b^	6283 (1176) ^a^
	Tissue density (g cm^−3^)	0.095 (0.004) ^a^	0.085 (0.015) ^a^	0.101 (0.036) ^a^	0.102 (0.022) ^a^

Different letters along the rows indicated significative differences among the means at *p* < 0.05 (test of Tukey). The values within the brackets are the error standards (n = 4).

**Table 5 plants-10-00005-t005:** Results of the PERMANOVA analysis testing the root morphology of maize plants against the stress treatment [Control (C), Drought (D), Heat (H) and Heat+Drought (HD)].

Factor	df	R^2^		F		*p*
Stress treatment	3	0.6693		8.0955		**<0.001**
Residual	12	0.3307				
Total	15	1.0000				
**Pairwise contrasts**						
	F model		R^2^		*p*	
C vs. D	9.262202		0.6068719		0.031	
C vs. H	3.877963		0.3925873		0.053	
C vs. HD	15.925573		0.7263469		0.027	
D vs. H	7.056074		0.5404438		0.034	
D vs. HD	13.677155		0.6950779		0.023	
H vs. HD	7.751958		0.5636985		0.054	

## Data Availability

The data presented in this study are available on request from the corresponding author.

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
