# Peer review of "Single and Combined Abiotic Stress in Maize Root Morphology"

_plants, 2020, doi:10.3390/plants10010005_

Round 1

Reviewer 1 Report

Your work on the phenotyping the response of seedlings of maize facing single and combined abiotic stresses (Heat and drought) is very interesting, with regard to the climate change. Introduction is clear and originality of the question is well presented.

You made a pretty good job, measuring many morphologic parameters on roots (and shoots) and use good combination of statistical analysis (ANOVA, PERMANOVA, Bansal & al method, sPLS-DA).
Could you precise if you measure the 5 plants /pot (x4 pots per stress) ? What about variation between pots ?

on the morphologic traits,  PFW and PDW have 2 different meanings : Plant fresh weight and Primary fresh weight (same with dry). I suggest you use different acronyms.

concerning equation(1), i did not find the signification of RF 

When presenting Bansal & al method, you used xCon for plant (line 313-plants experiencing combined stresses) then Comb for treatment (Line 314) ,then Comb plant (line 316, 318). It's confusing. Please clarify.

Regarding the tables and the results, I noticed several differences that should be clarified, for exemple line 74 to 77 / table 1 : (…) drougth stress significantly increased the RLR, (…) branching Density , but in table 1, branching Density is 4.961 (a) for control and 3.389 (c) for drougth, for me it's a decrease. Please check the sense of variation in text and in tables, and be cautious with the adjectif you used : table 1 versus text in lines 76-77 : (..) heat (..) weakly increased and reduced the fineness and branching Density, respectively (table1), but, in table 1, fineness with heat is 435(a) against 294(b) for control, for me it's stronger than "weakly", and branching Density is 4.5(a) for Heat against 2.8(b) for Control, for me it's NOT reduced. So please check. Our results are very interesting, they worth being presented in a clearer way. Also on al the tables, I suppose the number in brackets is the standard deviation ? you must write what this number in brackets means.

Figure 1 : add (A) and (B) on the plots ; Figure1(B) dry weigth : I"m surprised combined stress with a value of around 0.63 is 'c' when the values of the 'b' group are around 0.65 for heat and around  0.58 for drought. Could you check please ?

Figure 2 A versusu text : Braching zone on Figure 2A is additive, but on lines 103-105 it's said that  "the decrease of the branching zone Density in the primary roots under combined stress was the result of an antagonistics effet of the single stress;" please check the sentence, it's confusing, and not in cohérence with the figure.

figure 2C versus lines 108-109 : check the coherence please

figure 4 : sPLS-DA : I'm not convinced by the interpretation of component 3 : Heat convers a large part on composant 3, and it is partially superposed to others treatments on this component. Be careful in legend of figure 4 (B) :component 1 vs component 3 (not 2)

please explain a little bit more the contribution plots (figure 4 C,D, E) I'm not enought familiar with this graph and I do not understand what it brings.

I think your results are very interesting, and it's not obvious of being clear with this kind of data, but really it worth making an effort to highlight them. The question is important and your contribution is significant but be clearer.

Reviewer 2 Report

The results presented for evaluation are interesting. However, I would like to ask if it is possible to add a photographic documentation of the examined roots? In my opinion, this will definitely make this manuscript better. I am also asking for an explanation of the reasons for such large error bars in the individual figures, especially in the figures 2 and 3.

Reviewer 3 Report

The field environment often involves the simultaneous exposure of plants to more than one abiotic stress condition. The responses to stress combinations are unique, and cannot be predicted from separate analyses of the reactions to individual treatments. Hence physiological and morphological effects of combined stresses on plants has been extensively studied in the recent years. Heat stress, when combined with drought, is one of the major limitations to food production worldwide, especially in areas that use rainfed agriculture. Previous studies revealed that a combination of drought and heat stress might influence the growth and productivity of plants, however they were mainly focused on the aboveground plant parts. In the presented study the responses of the single root types of maize was evidenced. The primary and their lateral roots of this species were stimulated by the drought and heat combined stress with an increase of the length together with higher biomass allocation and fineness. Conversely, the length and biomass of the seminal lateral roots were sharply inhibited. These results have indicated the maize plants directed their internal resource towards to the root classes, such as the primary root and its laterals, that are mainly localized in the subsoil where could be present a water reserve. This outcome supports the view that root system architecture is an important developmental and agronomic trait, which plays vital roles in plant adaptation and productivity under stresses.

Reviewer 4 Report

Minor Comments

  1. The title of the study should be detailed and well organized with most appropriate words. Kindly rephrase the title of the study with more suitable scientific terminologies in an organized way.
  2. The abstract reflects the whole experimental work done, therefore, it should be free of grammatical mistakes. For the abstract part at line 12-14 of page 01 “Further, to face to the ……………. along the different organs” should be rephrased in an organized way, at line 19 of page 01 avoid the repetition/use of extra words such as “the” etc., at line 21-22 of page 01 the strategy suggested in the very last portion of abstract does not convey the message properly. Please elaborate it in more simple and appropriate words,
  3. Briefly mention the future aspects and the impact of the current study in the abstract section.
  4. For other sections the text should be proof read to rectify several mistakes like in introduction section at line 47 page 02 “differently responded differently…….” Avoid repetition and rectify the sentence, at line 50 of page 02 “… microcosm experiment were setup……” should be rephrased and rectified, at line 58-60 of page 02 “Hence, the maize…..… .……select informative variables” should be rephrased with more appropriate words, at line 62-63 of page 03 “It is highly productive under water suitable by irrigation…………..” should be organized. In results section at line 70 of page 02 “……both in terms of both fresh and dry weight”. Remove the repetition of words and rectify the sentence. In line 70 write “ significantly affected the plant growth in terms of fresh and dry weight” instead of “significantly affected the plant growth both in terms of both fresh and dry weight”。In the line 80 show that which effect the morphology;In line 81 write “Particularly” instead of “In particularly”.
  5. There are a lot of grammatical and spelling mistakes which directly influence the quality of the write-up. The overall write-up of the manuscript is very poor. Therefore, it has strongly recommended to improve the English and scientific write-up of the manuscript.
  6. Page No. 12, Figure 1 (B); the error bar for plant dry weight (g) while determining it for drought stress is missing. Kindly justify that.
  7. Page No.13, Figure 2; kindly make sure that you have provided all the necessary S.I units for the values used in these figures. If missing, please add it.
  8. Provide a list of abbreviations for all the short terminologies/words used in the article.
  9. Please provide the following photographs in supplementary data;
    1. Maize seedlings before harvesting (All treatments along controls).
    2. The morphological analysis of maize roots stained with 0.1 % toluidine blue solution under WinRHIZO image analysis (WinRhizo STD 1600, 281 Regent Instruments Inc., Canada).
  10. There are a lot of grammatical and spelling mistakes in this paper please correct it.
  11. Please provides the root photograph in the article.

Round 2

Reviewer 2 Report

Thank you very much, this manuscript is much better in its current form. Please accept it for printing.

Author Response

thanks for help for improving my manuscript